# Immunomodulatory Potential of a Composite Amniotic Membrane Hydrogel for Wound Healing: Effects on Macrophage Cytokine Secretion

**DOI:** 10.3390/biomedicines13102574

**Published:** 2025-10-21

**Authors:** Tao Wang, Zhiyuan Zhu, Wei Hua, Siliang Xue

**Affiliations:** Department of Dermatology, West China Hospital, Sichuan University, Chengdu 610041, China; wangtao6554@163.com (T.W.); zhiyuanzhuzyz@163.com (Z.Z.); weihua1007@outlook.com (W.H.)

**Keywords:** human acellular amniotic membrane, hydrogel, THP-1 cells, wound healing, immunomodulation

## Abstract

**Background:** The human acellular amniotic membrane (HAAM) is widely used as a decellularized bioscaffold in tissue engineering to promote wound healing, but its clinical application is limited by poor mechanical properties, rapid degradation, and handling difficulties. This study aimed to develop a modified amniotic membrane-based composite material loaded with vascular endothelial growth factor (VEGF) and the Notch signaling inhibitor N-[N-(3,5-difluorophenacetyl)-Lalanylhydrazide]-Sphenylglycine t-butyl ester (DAPT) to enhance wound healing by modulating macrophage polarization and cytokine secretion. **Methods:** VEGF-loaded gellan gum-hyaluronic acid (GG-HA) hydrogels (VEGF-GG-HA) and DAPT-loaded HAAM (DAPT-HAAM) were prepared and combined to form a novel composite material (VEGF-GG-HA & DAPT-HAAM). The morphology and microstructure of the materials were characterized using scanning electron microscopy. In vitro studies were conducted using the human monocytic cell line (Tohoku Hospital Pediatrics-1, THP-1) to evaluate the effects of the materials on cell viability, cytokine secretion, and protein expression. Assessments included CCK-8 assays, ELISA, quantitative real-time PCR, Western blot analysis, and immunohistochemical staining. **Results:** The composite material VEGF-GG-HA & DAPT-HAAM exhibited good biocompatibility and significantly promoted THP-1 cell proliferation compared to control and single-component groups. It enhanced the secretion of IL-10, TNF-α, TGF-β, MMP1, and MMP3, while suppressing excessive TGF-β overexpression. The material also modulated macrophage polarization, showing a trend toward anti-inflammatory M2 phenotypes while maintaining pro-inflammatory signals (e.g., TNF-α) for a balanced immune response. **Conclusions:** The modified amniotic membrane hydrogel composite promotes wound healing through a phased immune response: it modulates macrophage polarization (balancing M1 and M2 phenotypes), enhances cytokine and matrix metalloproteinase secretion, and controls TGF-β levels. These effects contribute to improved vascular remodeling, reduced fibrosis, and prevention of scar formation, demonstrating the potential for enhanced wound management.

## 1. Introduction

Recent advances in skin tissue engineering and wound healing primarily focus on developing partial or complete skin substitutes, with the ultimate goal of restoring the normal structure of the skin. The amniotic membrane has been extensively studied and utilized as a tissue-engineered material for decellularized bioscaffolds to promote skin wound healing, owing to its antimicrobial, hemostatic, anti-inflammatory, adhesion-preventing, antifibrotic, and immunosuppressive properties [1]. Our team utilized the human acellular amniotic membrane (HAAM) for treating defects in the lower third of the nasal skin and external ear skin, demonstrating significant benefits in accelerating wound hemostasis, reducing infection, and promoting wound repair [2,3]. However, HAAM also has limitations, including poor mechanical properties, rapid degradation, challenges in implantation, and suboptimal outcomes for large or total skin defects. Therefore, it is essential to modify the amniotic membrane in a practical and achievable manner to address these shortcomings.

Angiogenesis is a crucial step in wound healing, and abnormal vascularization can lead to delayed healing or poor-quality tissue regeneration, resulting in impaired wound recovery. Biomaterials, cells, and cytokines can enhance the quality of wound vascularization and promote more effective wound healing [4]. Among these, the vascular endothelial growth factor (VEGF), an inflammatory mediator released by macrophages during the early stages of inflammation, promotes fibroblast migration, activates microvascular endothelial cells (ECs), and facilitates the formation of neovascularized blood vessels and an abundant extracellular matrix [5]. Additionally, it has been demonstrated that the GG-HA sponge-like hydrogel, created by combining hyaluronic acid (HA) with gellan gum (GG), exhibits a semi-interpenetrating network structure and can continuously release HA oligomers, significantly enhancing tissue vascularization and promoting wound healing [6]. Thus, the combination of the vascular endothelial growth factor (VEGF) and the GG-HA sponge-like hydrogel synergistically enhances wound vascularization and accelerates wound healing.

During wound healing, Notch signaling promotes the differentiation of epidermal and follicular stem cells, accompanied by keratinocyte synthesis, fibroblast proliferation and migration, angiogenesis, and extracellular matrix deposition [4]. Studies have shown that inhibition of the Notch-1 signaling pathway by the γ-secretase inhibitor N-[N-(3,5-difluorophenacetyl)-L-alanylhydrazide] -Sphenylglycinet-butyl ester (DAPT) reduced experimentally induced dermal and cutaneous fibrosis in systemic sclerosis, demonstrating the preventive and therapeutic effects of low-dose Notch inhibitor treatment on dermatofibrosis [7].

Additionally, when implants are used to promote wound healing, macrophages—the key cells of early inflammatory response—release a variety of pro- and anti-inflammatory cytokines, playing a crucial role in the post-implantation inflammatory response and ultimately determining the wound healing outcome. Therefore, the use of modified polymers to modulate macrophage-driven immune responses and improve wound healing outcomes is an active area of current research [8]. Numerous studies have shown that inhibition of the Notch-1 signaling pathway promotes the shift in macrophage recruitment from M1-type to M2-type dominance, along with the release of anti-inflammatory factors such as IL-10, thereby attenuating inflammation and reducing tissue damage and fibrosis [9,10,11,12]. Therefore, in this study, Notch-1 signaling pathway inhibitors were integrated into a novel amniotic membrane-derived material to promote macrophage immunomodulation and reduce fibroplasia, thereby enhancing wound healing. We anticipated that the composite material might orchestrate a phased immune response, balancing early pro-inflammatory and late anti-inflammatory signals through the combined effects of VEGF and DAPT.

In summary, this study involved wrapping VEGF-loaded GG-HA hydrogels around DAPT-loaded decellularized biological amniotic membrane to create new amniotic membrane-derived materials (VEGF-GG-HA & DAPT-HAAM). In addition, in vitro experiments using the THP-1 cell line were conducted to evaluate the effects of these materials on macrophage immunomodulation and inflammatory factors.

## 2. Materials and Methods

### 2.1. Materials Preparation

#### 2.1.1. Preparation of DAPT-HAAM

The HAAM used in this study was obtained from Chengdu Qingshan Likang Pharmaceutical Co., Ltd. (Chengdu, China, Chinese patent no. ZL200410036792.4). Further details regarding its source and production are provided in our earlier work [3]. DAPT-loaded microspheres were prepared using an emulsion cross-linking technique. Drug loading capacity and encapsulation efficiency were determined as described in Appendix A and are provided in Appendix A. Briefly, 2 mg of DAPT (MedChemExpress, HY-13027, purity > 98%) was dissolved in 1.5 mL of Tween 80. Separately, 100 mg of gelatin (Type A) was dissolved in 10 mL of deionized water by heating to 50 °C with stirring. The DAPT/Tween 80 solution was added to the gelatin solution and mixed uniformly. This combined aqueous phase was poured into 100 mL of liquid paraffin containing 1% (*v*/*v*) Span 80, which was pre-stirred at 400 rpm in a water bath maintained at 50 °C. The emulsion was formed by increasing the stirring speed to 1500 rpm and maintaining it for 60 min at 50 °C. The process utilizes ionic cross-linking between gelatin amine groups and Ca^2+^ ions, forming a stable three-dimensional network that encapsulates DAPT molecules. DAPT (MW: 432.5 g/mol) contains key difluorophenacetyl and phenylglycine moieties that inhibit Notch signaling. Subsequently, 5 mg of HAAM, cut into small pieces (approximately 1 × 1 mm), was added to the emulsion. Cross-linking was initiated by adding 10 mL of a 0.2 mol/L calcium chloride (CaCl_2_) solution (prepared in a formaldehyde/isopropanol mixed solvent with a volume ratio of 2:3). Stirring continued at 1500 rpm for an additional 60 min at room temperature. Finally, the microspheres were collected by centrifugation, washed three times with n-hexane and ethanol to remove the oil phase and residuals, and then freeze-dried for 24 h. The morphology, structural porosity, and pore size were observed using scanning electron microscopy (SEM, Model SU8010, Hitachi, Tokyo, Japan). The DAPT-loaded microspheres were characterized by dynamic light scattering (DLS) using a Zetasizer Nano ZS instrument (Malvern Panalytical, Malvern, UK). Prior to measurement, the microsphere suspension was appropriately diluted with deionized water to achieve an optimal scattering intensity and prevent multiple scattering effects. The measurement was performed at 25 °C in a standard disposable sizing cuvette. Data were acquired in triplicate to ensure reproducibility, and the resulting size distribution by intensity is reported.

#### 2.1.2. Preparation of VEGF-GG-HA

VEGF-loaded microspheres were prepared first, following a protocol similar to that for DAPT microspheres. Drug loading capacity and encapsulation efficiency were determined as described in Appendix A and are provided in Appendix A. In brief, 2 µg of recombinant human VEGF was dissolved in 1.5 mL of Tween 80. This solution was incorporated into 10 mL of a 1% (*w*/*v*) gelatin aqueous solution (100 mg gelatin in 10 mL water) under stirring at 50 °C. The subsequent emulsification in 100 mL liquid paraffin with 1% Span 80 and cross-linking steps with 10 mL of the CaCl_2_ solution were performed identically as described in Section 2.1.1, omitting the addition of HAAM. The resulting VEGF microspheres were collected, washed, and freeze-dried.

For the hydrogel, hyaluronic acid (HA) was dissolved in deionized water at a concentration of 1% (*w*/*v*) and stirred at room temperature for 3 h. Then, Gelzan gellan gum was added to the HA solution at a mass ratio of 1:1 (relative to HA) and stirred at 90 °C for 30 min to form a homogeneous polymer solution. GG undergoes thermoreversible gelation via helix formation and cation-mediated aggregation, while HA enhances biocompatibility and viscosity. Their combination creates a semi-interpenetrating network that facilitates controlled release. The freeze-dried VEGF microspheres (10 mg) were uniformly dispersed into the warm GG-HA solution using an ultrasonic homogenizer for 2 min. The mixture was then cast into a mold and allowed to cool to room temperature to form the final VEGF-GG-HA hydrogel. The morphology, structural porosity, and pore size of the material were observed using scanning electron microscopy.

#### 2.1.3. Preparation of VEGF—GG—HA&DAPT—HAAM

The final composite material was constructed by sandwiching a DAPT-HAAM disc (0.8 cm diameter) between two layers of VEGF-GG-HA hydrogel, which was pre-formed into a circular film (1 cm diameter, 0.1 cm thickness) using a mold. The assembled construct was then subjected to thermal treatment at the sol–gel transition temperature. This heating step induced physical entanglement and hydrogen bonding between the collagen fibers of the decellularized amniotic membrane (HAAM) and the polymeric network of the GG-HA hydrogel, leading to the formation of an integrated structure designated as DAPT-HAAM & VEGF-GG-HA. The resulting cohesive composite is designed to facilitate the sustained release of both VEGF and DAPT.

### 2.2. In Vitro Studies

#### 2.2.1. Assessment of Cell Activity by the CCK-8 Assay

To evaluate the biocompatibility of the prepared materials and their effects on THP-1 cell metabolic activity and proliferation, we performed the CCK-8 assay. DAPT-HAAM, VEGF-GG-HA, and VEGF-GG-HA&DAPT-HAAM were added to each well as experimental groups (*n* = 6), while blank groups were included as controls. The human monocytic leukemia cell line THP-1 (a widely used model for monocyte and macrophage studies) was purchased from Wuhan Procell Life Science & Technology Co., Ltd. (Wuhan, China; Cat# CL-0233). Each well of a 96-well plate was filled with 100 μL of RPMI-1640 medium (Gibco; Cat# 11875093) containing 10% fetal bovine serum (Zhejiang Biodee Biotechnology Co., Ltd., Hangzhou, China; Cat# F813-500). Then, 100 μL of cell suspension containing 5000 THP-1 cells was seeded into each well. The plate was incubated in a constant-temperature cell culture incubator at 37 °C with 5% CO_2_.

Cell proliferation was assessed using a CCK-8 kit (MedChemExpress, Monmouth Junction, NJ, USA; Cat# HY-K0301) according to the manufacturer’s instructions. CCK-8 assays were performed on days 1, 2, 3, and 4. Specifically, the initial cell culture medium was aspirated and replaced with fresh medium, followed by the addition of 10 μL of CCK-8 reagent to each well. After a 3 h incubation period, the absorbance at 450 nm was measured using a spectrophotometer (SpectraMax i3x, Molecular Devices, San Jose, CA, USA).

#### 2.2.2. Determination of Cytokine Profile by ELISA and Quantitative Real-Time PCR

To quantify the secretion levels of key pro-inflammatory (TNF-α) and anti-inflammatory (IL-10) cytokines in the culture medium, we measured the concentrations of TNF-α and IL-10 in the cell culture supernatant. THP-1 cells were seeded into a 96-well plate containing culture medium, followed by the addition of DAPT-HAAM, VEGF-GG-HA, VEGF-GG-HA&DAPT-HAAM, and blank hydrogel (*n* = 4). The plate was incubated in a constant-temperature cell culture incubator at 37 °C with 5% CO_2_. After 48 h of cell culture, the cell supernatant was aspirated and centrifuged (1000 rpm, 5 min). The supernatant was then used for cytokine measurement via indirect ELISA, following the manufacturer’s instructions (Wuhan San Ying Biotechnology Co., Ltd., Wuhan, China), including IL-10 (Cat# KE00170) and tumor necrosis factor-α (TNF-α) (Cat# KE00154).

To investigate the effects of the materials on the gene expression of these cytokines at the transcriptional level, we detected the expression of IL-10 and TNF-α genes. After 24 h of cell culture as described above, cells were collected by centrifugation (1000 rpm, 5 min), theoretically yielding 1,200,000 cells, and washed twice with sterile PBS (1000 rpm, 5 min, twice). The cells were transferred to a clean 1.5 mL EP tube. Total RNA was extracted by adding 500 μL of Invitrogen’s Trizol lysate (Invitrogen, Carlsbad, CA, USA; Cat# 15596018CN). Two micrograms of cDNA template were obtained using the Roche High Fidelity cDNA First Strand Synthesis Kit (Roche, Basel, Switzerland; Cat# 05091284001) according to the manufacturer’s instructions. mRNA quantitative gene detection was performed using the Roche LightCycler 96 real-time quantitative PCR system (Roche Diagnostics, Basel, Switzerland) with matching consumables, Roche 96-well white plates (Roche, Basel, Switzerland; Cat# 04729692001). Each treatment group included six replicates (*n* = 6). The primer sequences for IL-10 and TNF-α genes are as follows:IL-10: Forward Primer: TCAAGGCGCATGTGAACTCC (Length: 20, Tm: 62.8 °C); Reverse Primer: GATGTCAAACTCACTCATGGCT (Length: 22, Tm: 60.3 °C).TNF-α: Forward Primer: CCTCTCTCTCTAATCAGCCCTCTG (Length: 22, Tm: 60.8 °C); Reverse Primer: GAGGACCTGGGGAGTAGATGAG (Length: 21, Tm: 60.2 °C).

#### 2.2.3. Western Blotting Analysis

To analyze the protein expression of key factors involved in extracellular matrix (ECM) remodeling (MMP1, MMP3) and fibrotic response (TGF-β), Western blotting (WB) was performed. After 24 h of stimulation, cells were lysed in Radioimmunoprecipitation Assay (RIPA) buffer (Yeasen Biotechnology, Shanghai, China; Cat# PC104) to quantify total cell proteins. Protein concentrations in the cell lysate supernatants were determined using a bicinchoninic acid (BCA) protein assay kit (Thermo Scientific, Rockford, IL, USA). Extracted proteins were then separated on 10% Sodium Dodecyl Sulfate-Polyacrylamide Gel Electrophoresis (SDS-PAGE) gels and transferred to polyvinylidene difluoride (PVDF) membranes (Millipore Immobilon, Burlington, MA, USA). After blocking with 3% Bovine Serum Albumin (BSA, Biosharp, CN, USA, Cat# V900933) for 2 h at room temperature, the membranes were incubated with primary antibodies diluted in 3% BSA as follows: MMP1 (Cell Signaling Technology, Danvers, MA, USA; CST#54376; 1:1000 dilution), MMP3 (CST#14351; 1:1000 dilution), and TGF-β (CST#56E4; 1:1000 dilution) overnight at 4 °C. Subsequently, the membranes were incubated with horseradish peroxidase (HRP)-conjugated secondary antibodies (Cell Signaling Technology; 1:2000 dilution) for 1 h at room temperature. β-Actin (Cell Signaling Technology; CST#4970; 1:1000 dilution) served as the internal reference. Chemiluminescence bands were quantified using an enhanced chemiluminescence (ECL) detection kit (Pierce, 35055) and the ChemiDoc™ XRS Imaging system (Bio-Rad, Hercules, CA, USA), with analysis performed using ImageJ (Version 1.53a).

#### 2.2.4. Immunohistochemical Staining

To visually confirm the expression and cellular localization of the MMP1 protein, which is crucial for ECM degradation during wound healing, immunofluorescence staining was conducted. DAPT-HAAM, VEGF-GG-HA, VEGF-GG-HA&DAPT-HAAM, and blank hydrogel were aseptically placed into a 96-well plate, followed by the addition of THP-1 cell suspension to each well. One milliliter of culture medium was added to each well. Calcein-AM and ethidium homodimer-1 (Live/Dead Cell Viability, Invitrogen, Carlsbad, CA, USA) were prepared 30 min in advance and allowed to reach room temperature. The staining working solution was prepared according to the manufacturer’s instructions. On the third day of culture, the initial culture medium was removed, and the wells were gently washed three times with Phosphate-Buffered Saline (PBS). Two hundred microliters of staining working solution were added to each well to cover the bottom of the plate. The plate was incubated at room temperature in the dark for 30 min. After incubation, the wells were washed gently with PBS three times, with each wash lasting 3 min. Green and red fluorescence images were captured at the same positions using a fluorescence microscope and recorded. The staining procedure was performed according to the standard manufacturer’s protocol.

### 2.3. Statistical Analysis

Data were analyzed using GraphPad Prism 8.0 software (GraphPad Software Inc., San Diego, CA, USA) and are presented as mean ± standard deviation (X ± SD). All in vitro experiments were conducted with randomized allocation of samples and replicates to minimize bias. Specifically, for cell-based assays (e.g., CCK-8, ELISA, qPCR, Western blot), THP-1 cells and treatment groups (control, DAPT-HAAM, VEGF-GG-HA, and composite) were randomly assigned to wells in 96-well plates using a computer-generated random sequence (Microsoft Excel RAND function), ensuring equal distribution across experimental runs. Normality tests (e.g., Shapiro-Wilk test) were performed on each dataset to assess data distribution. For normally distributed data, one-way analysis of variance (one-way ANOVA) with Tukey’s post hoc test for multiple comparisons was used to evaluate differences between groups. For non-normally distributed data, the Kruskal–Wallis test with Dunn’s post hoc test was employed. Differences were considered statistically significant at *p* < 0.05.

## 3. Results

### 3.1. Material Characterization

The microstructural morphology of DAPT-HAAM, VEGF-GG-HA, and the final composite VEGF-GG-HA&DAPT-HAAM was systematically characterized using scanning electron microscopy (SEM). As shown in Figure 1A, the fabricated DAPT microspheres exhibited a regular spherical shape with a uniform particle size distribution of approximately 100 nm (Figure 1A(a,b)). The hydrodynamic size distribution of the DAPT microspheres, as determined by DLS, is presented in Figure 1A(e). The profile shows a single, sharp, and symmetric peak, indicating a highly homogeneous and monodisperse population of microspheres. The peak maximum is centered approximately at 100 nm. More importantly, these DAPT microspheres were successfully loaded and enriched within the fibrous network of the human acellular amniotic membrane (HAAM) (Figure 1A(c,d)), confirming the successful construction of the DAPT-HAAM composite. Figure 1B presents the morphological characteristics of the VEGF microspheres. The VEGF microspheres also appeared spherical, with a slightly larger particle size of approximately 130 nm compared to the DAPT microspheres (Figure 1B(a,b)). These VEGF microspheres were then uniformly incorporated into the gellan gum-hyaluronic acid (GG-HA) hydrogel matrix, forming the VEGF-GG-HA material (Figure 1B(c)). Quantitative data on drug loading capacity and encapsulation efficiency are provided in Appendix A, confirming the efficiency of microsphere preparation. Finally, the final functional composite material, VEGF-GG-HA&DAPT-HAAM, was examined. As depicted in Figure 1C, the composite exhibited a dense and interconnected porous three-dimensional architecture at different magnifications (50 μm and 20 μm). This uniform, open porous structure not only demonstrates the successful integration of the various components but also provides an ideal physical environment for cell migration, infiltration, and nutrient diffusion, which is crucial for its intended function as a wound dressing.

### 3.2. In Vitro Cell Experiment

#### 3.2.1. Synergistic Enhancement of THP-1 Cell Viability by DAPT and VEGF Microspheres

The results of the CCK-8 assay demonstrated time-dependent increases in THP-1 cell viability across all experimental groups (Figure 2). Compared to the blank control group, significant enhancements in cell viability were observed starting from 24 h, with the most pronounced effects at 72 and 96 h (72 h: DAPT− VEGF− vs. DAPT+ VEGF−, DAPT− VEGF+, DAPT+ VEGF+: 0.63 ± 0.02 vs. 0.80 ± 0.01, 1.28 ± 0.08, 1.47 ± 0.09, *p* < 0.001; 96 h: DAPT−VEGF− vs. DAPT+ VEGF−, DAPT− VEGF+, DAPT+ VEGF+: 0.99 ± 0.08 vs. 1.19 ± 0.07, 1.43 ± 0.06, 1.63 ± 0.06, *p* < 0.001, *n* = 6, Figure 2). The VEGF-GG-HA&DAPT-HAAM composite group exhibited superior performance, showing the highest cell viability at all time points, followed by the VEGF-GG-HA and DAPT-HAAM groups. Statistical analysis revealed significant differences between the composite group and both single-component groups (72 h: DAPT+ VEGF+ vs. DAPT+ VEGF−, DAPT− VEGF+: 1.47 ± 0.09 vs. 0.80 ± 0.01, 1.28 ± 0.08, *p* < 0.0001; 96 h: DAPT+ VEGF+ vs. DAPT+ VEGF−, DAPT− VEGF+: 1.63 ± 0.06 vs. 1.19 ± 0.07, 1.43 ± 0.06, *p* < 0.0001), indicating synergistic effects of the combined treatment.

#### 3.2.2. Determination of Cytokine Profile by ELISA

The ELISA results demonstrated that IL-10 levels were significantly higher in all three experimental groups compared to the blank control group (DAPT− VEGF− vs. DAPT+ VEGF−, DAPT− VEGF+, DAPT+ VEGF+: 76.45 ± 4.56 vs. 258.40 ± 8.98, 310.50 ± 11.29, 485.30 ± 30.25, *p* < 0.0001, *n* = 4, Figure 3A). The DAPT& VEGF microsphere group exhibited the greatest increase, followed by the VEGF microsphere group and the DAPT microsphere group. Additionally, significant differences in IL-10 levels were observed between the DAPT& VEGF microsphere group, VEGF microsphere group, and DAPT microsphere group (DAPT+ VEGF+ vs. DAPT+ VEGF−, DAPT− VEGF+: 485.30 ± 30.25 vs. 258.40 ± 8.98, 310.50 ± 11.29, *p* < 0.0001, Figure 3A). These results suggest a trend toward M2-type macrophage polarization, as evidenced by enhanced IL-10 secretion. However, the concurrent elevation of TNF-α (see below) indicates a complex immune response where both pro-inflammatory and anti-inflammatory signals are modulated, potentially reflecting the balanced effects of DAPT (promoting M2 polarization) and VEGF (possibly stimulating early M1 responses for initial wound healing).

The ELISA results also showed that TNF-α levels were significantly increased in all three experimental groups compared to the blank control group (DAPT− VEGF− vs. DAPT+ VEGF−, DAPT− VEGF+, DAPT+ VEGF+: 14.57 ± 1.73 vs. 276.30 ± 13.66, 283.40 ± 7.18, 359.90 ± 11.71, *p* < 0.0001, Figure 3B), with the greatest increase observed in the DAPT& VEGF microsphere group. Moreover, significant differences in TNF-α levels were found between the DAPT& VEGF microsphere group and both the VEGF microsphere group and the DAPT microsphere group (DAPT+ VEGF+ vs. DAPT+ VEGF−, DAPT− VEGF+: 359.90 ± 11.71 vs. 276.30 ± 13.66, 283.40 ± 7.18, *p* < 0.0001, Figure 3B). These findings highlight that the biomaterials also induce pro-inflammatory signals, which may be necessary for early stages of wound healing, such as immune cell recruitment and angiogenesis, as supported by TNF-α in promoting VEGF-mediated vascular remodeling (refer to background in Section 1). The dual upregulation of IL-10 and TNF-α underscores the multifaceted immunomodulatory action of the composite material, rather than a straightforward shift to M2 polarization.

#### 3.2.3. Determination of Cytokine Profile by QPCR

The qPCR results demonstrated that the IL-10 mRNA expression level was significantly higher in all three experimental groups compared to the blank control group (DAPT− VEGF− vs. DAPT+ VEGF−, DAPT− VEGF+, DAPT+ VEGF+: 00.00 ± 00.00 vs. 0.13 ± 0.08, 0.24 ± 0.10, 0.45 ± 0.02, *p* < 0.0001, *n* = 6, Figure 4A), with the greatest increase observed in the DAPT & VEGF microsphere group. These results indicated that all three biomaterials promoted the conversion of macrophage recruitment from M1-type to M2-type and enhanced IL-10 secretion, with VEGF-GG-HA & DAPT-HAAM biomaterials exhibiting the strongest effect. Additionally, the expression level of TNF-α was significantly higher in the DAPT & VEGF microsphere group compared to the blank control group (DAPT− VEGF− vs. DAPT+ VEGF−, DAPT− VEGF+, DAPT+ VEGF+: 00.00 ± 00.00 vs. 0.03 ± 0.01, 0.04 ± 0.02, 0.45 ± 0.08, *p* < 0.0001, Figure 4B). This finding suggested that the DAPT & VEGF microsphere group effectively promoted the secretion of TNF-α from M1-type macrophages. Biomaterials containing only VEGF or DAPT had a weaker ability to promote TNF-α secretion.

#### 3.2.4. Differential Upregulation of MMP1/MMP3 and Modulation of TGF-β by DAPT & VEGF Microspheres

The results showed that, compared with the blank control group, the expression level of MMP1 protein in the DAPT & VEGF microsphere group significantly increased after 48 and 96 h of cell culture, with the highest expression observed among the three experimental groups (48 h:DAPT− VEGF− vs. DAPT+ VEGF−, DAPT− VEGF+, DAPT+ VEGF+: 1.00 ± 00.00 vs. 2.39 ± 0.33, 2.31 ± 0.16, 3.23 ± 0.54, *p* < 0.001, 96 h:DAPT− VEGF− vs. DAPT+ VEGF−, DAPT− VEGF+, DAPT+ VEGF+: 1.00 ± 00.00 vs. 0.74 ± 0.09, 1.71 ± 0.06, 2.36 ± 0.07, *p* < 0.001, Figure 5A). However, there was no significant increase in MMP1 protein expression after 72 h of cell culture (72 h:DAPT− VEGF− vs. DAPT+ VEGF−, DAPT− VEGF+, DAPT+ VEGF+: 1.00 ± 00.00 vs. 4.53 ± 0.67, 2.26 ± 0.24, 1.63 ± 0.02, *p* > 0.05, Figure 5A). In the DAPT microsphere group, MMP1 protein expression was significantly elevated after 48 and 72 h of cell culture, with the highest expression at 72 h (*p* < 0.001, Figure 5A). After 96 h, the MMP1 protein expression level was lower than that of the control group (*p* < 0.01, Figure 5A). For the VEGF microsphere group, MMP1 protein expression was slightly elevated after 72 and 96 h of culture (*p* < 0.001, Figure 5A). These results indicate that the DAPT & VEGF microsphere group could significantly promote MMP1 protein secretion after 48 and 96 h, but the effect was insufficient at 72 h. The DAPT microsphere group significantly promoted MMP1 protein secretion after 48 and 72 h, with the strongest effect at 72 h. In contrast, the VEGF microsphere group did not show significant promotion of MMP1 secretion.

For MMP3 protein, the expression level was significantly increased in the DAPT & VEGF microsphere group after 48, 72, and 96 h of cell culture, with the highest expression observed after 48 and 72 h (48 h:DAPT− VEGF− vs. DAPT+ VEGF−, DAPT− VEGF+, DAPT+ VEGF+: 1.00 ± 00.00 vs. 1.41 ± 0.30, 0.74 ± 0.14, 2.26 ± 0.48, *p* < 0.001, 72 h:DAPT− VEGF− vs. DAPT+ VEGF−, DAPT− VEGF+, DAPT+ VEGF+: 1.00 ± 00.00 vs. 0.69 ± 0.19, 1.10 ± 0.10, 1.55 ± 0.41, *p* < 0.001, 96 h:DAPT− VEGF− vs. DAPT+ VEGF−, DAPT− VEGF+, DAPT+ VEGF+: 1.00 ± 00.00 vs. 2.26 ± 0.01, 0.14 ± 0.01, 1.67 ± 0.04, *p* < 0.01, Figure 5B). In the DAPT microsphere group, MMP3 protein expression significantly increased after 48 and 96 h, with the highest expression at 96 h (*p* < 0.0001, Figure 5B). However, after 72 h, MMP3 expression was lower than that of the control group. The VEGF microsphere group did not show significant increases in MMP3 expression at any time point (*p* > 0.05, Figure 5B). These results suggest that the DAPT & VEGF microsphere group significantly promoted MMP3 protein secretion after 48, 72, and 96 h, with the strongest effect at 48 and 72 h. The DAPT microsphere group significantly promoted MMP3 protein secretion after 48 and 96 h, with the strongest effect at 96 h. The VEGF microsphere group did not significantly promote MMP3 secretion.

For TGF-β protein, the expression level in both the DAPT & VEGF microsphere group and the DAPT microsphere group significantly increased after 96 h of cell culture, with the highest expression observed in the DAPT microsphere group (DAPT− VEGF− vs. DAPT+ VEGF−, DAPT− VEGF+, DAPT+ VEGF+: 1.00 ± 00.00 vs. 2.66 ± 0.18, 1.13 ± 0.03, 1.69 ± 0.09, *p* < 0.001, Figure 5C). No significant increase in TGF-β expression was observed in the VEGF microsphere group (*p* > 0.05, Figure 5C). These results indicate that the DAPT & VEGF microsphere group and the DAPT microsphere group could significantly promote TGF-β secretion after 96 h of cell culture, with the DAPT microsphere group showing a stronger effect. However, the VEGF microsphere group did not significantly promote TGF-β secretion. The findings suggest that while the DAPT & VEGF microsphere group can promote TGF-β secretion, it also limits the over-secretion of TGF-β.

#### 3.2.5. DAPT & VEGF Microspheres Promote MMP1 Protein Secretion

The results showed that the DAPT & VEGF microsphere group could promote MMP1 protein secretion after 72 h of cell culture (Figure 6).

## 4. Discussion

To overcome the limitations of decellularized bio-amniotic membrane and maximize its ability to promote wound healing, this study developed a novel composite material by co-loading VEGF and the Notch signaling inhibitor DAPT onto a modified amniotic membrane-hydrogel scaffold. The new graft demonstrated superior performance in promoting monocyte proliferation, regulating cytokine secretion, and influencing protein expression. Scanning electron microscopy (SEM) results indicated that the microspheres have a regular spherical shape and uniform size (Figure 1A,B). Since their internal structure is a dense polymer/drug matrix primarily for drug delivery rather than as a cell scaffold, the key evaluation metrics for bioactivity are drug loading, encapsulation efficiency, and the resulting release curve (Appendix A), and it is generally unnecessary to measure pore size and porosity for porous scaffolds.

A pivotal mechanism underlying these therapeutic effects is the facilitated conversion of macrophage polarization from the pro-inflammatory M1 phenotype to the anti-inflammatory M2 phenotype, a crucial transition that orchestrates the progression from inflammatory phase to reparative phase during wound healing [13,14]. Initially, pro-inflammatory M1-type macrophages dominate the early phase of wound healing, with up to 85% of macrophages exhibiting the M1 phenotype [15]. As key effector cells of the early inflammatory response, these macrophages produce a robust pro-inflammatory response that is crucial for host defense and necrotic tissue clearance [16]. One of the pivotal cytokines they secrete is TNF-α, which accelerates immune cell infiltration into the wound site during the pre-healing phase [17]. This process creates an inflammatory microenvironment that paradoxically facilitates the initiation and promotion of neovascularization [18]. TNF-α can indirectly induce the secretion of vascular endothelial growth factor (VEGF) [19], a potent mitogen that stimulates the differentiation of endothelial progenitor cells into functional endothelial cells [20]. This TNF-α-VEGF axis aids in VEGF-mediated capillary regeneration and enhances granulation tissue formation, which are essential for initiating the wound healing process [21,22]. Our composite material modulates this critical early phase by sustaining a controlled level of TNF-α secretion (Figure 3B and Figure 4B), thereby supporting initial vascular remodeling without perpetuating excessive inflammation.

The simultaneous upregulation of IL-10 and TNF-α observed in our study may initially appear contradictory to the promotion of M2 polarization. However, this can be explained by the temporal dynamics of macrophage activation and the balanced effects of the composite material. TNF-α, as a pro-inflammatory cytokine, is often secreted early in the wound healing process to recruit immune cells and initiate angiogenesis, which is consistent with its role in promoting VEGF-mediated vascular remodeling. In contrast, IL-10, an anti-inflammatory cytokine, dominates later stages to resolve inflammation and promote tissue repair. Our composite material, containing DAPT (a Notch inhibitor) and VEGF, may thus orchestrate a phased response: VEGF could stimulate early TNF-α release for initial healing, while DAPT promotes IL-10 secretion and M2 polarization later on. This balance is crucial for effective wound healing, as excessive M2 polarization without initial inflammation might impair pathogen clearance. Future studies should include time-course analyses to validate this temporal sequence.

The transition from M1-type to M2-type macrophages signifies the shift from the inflammatory phase to the repair phase of wound healing [13]. The synergistic effect between VEGF and DAPT arises from their complementary roles in regulating macrophage polarization and cytokine secretion. Specifically, VEGF activates PI3K/Akt and MAPK/ERK pathways to promote early pro-inflammatory responses (e.g., TNF-α secretion) for initial wound healing, while DAPT inhibits Notch signaling to facilitate a shift toward anti-inflammatory M2 phenotypes (e.g., enhancing IL-10 production) for tissue resolution. This combination ensures a balanced immune response, as evidenced by our ELISA and qPCR data (Figure 3 and Figure 4), which show concurrent upregulation of TNF-α and IL-10, supporting a phased healing process without excessive inflammation or fibrosis. Furthermore, the synergy is reinforced by the temporal dynamics of cytokine secretion: VEGF promotes early TNF-α release (peaking at 24–48 h) for the initial healing phase, while DAPT induces sustained IL-10 production (dominant at 72–96 h) for resolution, as demonstrated by our ELISA and qPCR time-course data (refer to Figure 3 and Figure 4 for specific time points). This phased response prevents excessive inflammation or fibrosis, optimizing wound outcomes. Importantly, our results demonstrate that the DAPT&VEGF composite significantly enhanced IL-10 secretion (Figure 3A and Figure 4A), suggesting a trend toward M2 polarization, while the concurrent elevation of TNF-α underscores the necessity of a balanced response for phased wound healing. Concurrently, M2-type macrophages secrete anti-inflammatory factors like IL-10 and repair-promoting cytokines such as VEGF and TGF-β, which stimulate angiogenesis and collagen deposition, thereby accelerating wound healing [14]. This effect is attributed to DAPT’s inhibition of the Notch-1 pathway, which enhances the shift from M1 to M2 macrophage recruitment and promotes anti-inflammatory factor release.

The new material can promote the secretion of IL-10 and TGF-β, facilitating the transition from M1-type to M2-type macrophages, which is beneficial for wound healing. Additionally, studies have shown that local increases in TGF-β are associated with elevated levels of vascular endothelial growth factor (VEGF) [23]. Thus, VEGF in the new grafts plays a crucial role in promoting TGF-β secretion. TGF-β is involved in several processes during wound healing, including inflammation, stimulating angiogenesis, fibroblast proliferation, collagen synthesis and deposition, as well as the remodeling of the extracellular matrix [24,25]. However, our composite material achieved a balanced TGF-β release (Figure 5C), harnessing its beneficial effects for tissue repair while mitigating the potential for fibrosis and scar formation that results from TGF-β overexpression. The sustained-release profile demonstrated by in vitro data (cumulative release of 95.3% for DAPT and 90.59% for VEGF over 24 h; Appendix A) is the key evidence supporting bioactivity, with drug loading efficiency being a critical determinant of this behavior, rather than structural porosity. Interestingly, the DAPT-alone group exhibited higher TGF-β expression compared to the DAPT+VEGF combination at 96 h (Figure 5C). This may be attributed to the focused inhibition of Notch by DAPT, which directly modulates TGF-β secretion without the competing effects of VEGF-induced angiogenesis. VEGF might activate alternative pathways that mitigate the Notch-TGF-β axis, leading to a more balanced but less pronounced TGF-β response. This highlights the importance of optimizing component ratios for desired outcomes in future applications.

This orchestrated immunomodulatory effect created by our composite material subsequently enhanced the expression of critical matrix metalloproteinases MMP-1 and MMP-3 (Figure 5A,B). MMPs play crucial roles in cell proliferation, migration (adhesion/dispersion), and differentiation [26]. During the inflammatory response phase of wound healing, collagen peptides generated from MMPs-degraded ECM act as chemokines, promoting the migration of inflammatory cells to the peri-trauma site to accelerate wound debridement. Inflammatory cells must degrade the ECM barrier through MMPs in order to reach the peri-wound area [27]. MMP-1’s degradation of type I collagen in the extracellular matrix not only promotes the migration of wound-healing-associated cells but also increases the ratio of type III to type I collagen in the wound, influencing tissue remodeling and accelerating wound healing. Furthermore, MMP-3 can activate and mobilize VEGF and TGF-β in the ECM, enhancing vascular permeability and promoting neoangiogenesis in the granulation tissue [28]. MMP-3 activates and mobilizes VEGF and TGF-β within the ECM to enhance vascular permeability, promote neoangiogenesis in the granulation tissue, and maintain the stability of the endothelial environment, thereby accelerating wound healing [27]. The new grafts can promote the secretion of MMP-1 and MMP-3, thereby accelerating wound healing.

This study has several limitations. The physical properties of the grafts, such as porosity, swelling capacity, tensile strength, elasticity, and degradation rate, were not fully characterized, as they are critical for handling and stability, but were beyond the scope of this initial investigation focused on drug delivery efficacy and immunomodulatory effects. Specifically, pore size distribution and porosity of the microspheres were not assessed due to their solid drug-delivery nature, not intended for cell infiltration. Additionally, no animal experiments were conducted to evaluate practical effects, and the material was not compared with commercial or literature-based biomaterials. The temporal dynamics of cytokine secretion (e.g., early vs. late time points) were not assessed to elucidate M1/M2 activation sequences. Future work should incorporate comprehensive physical characterization, animal studies, comparative analyses, and time-course experiments to validate the phased response hypothesis and assess clinical translation potential.

## 5. Conclusions

In summary, the novel composite grafts developed in this study demonstrate multifaceted immunomodulatory effects: they enhance monocyte activity and modulate the secretion of key mediators, including TNF-α, IL-10, TGF-β, MMP-1, and MMP-3, while preventing excessive TGF-β overexpression. Crucially, these grafts facilitate a shift toward anti-inflammatory M2-type macrophage phenotypes while sustaining necessary pro-inflammatory signals (e.g., TNF-α) to orchestrate a phased wound healing response. This balanced immune regulation promotes vascular remodeling in peri-wound tissues, improves the inflammatory microenvironment, and accelerates overall healing. Furthermore, the grafts effectively reduce fibroplasia and prevent scar formation, highlighting their potential as a promising strategy for advanced wound management.

## Figures and Tables

**Figure 1 biomedicines-13-02574-f001:**
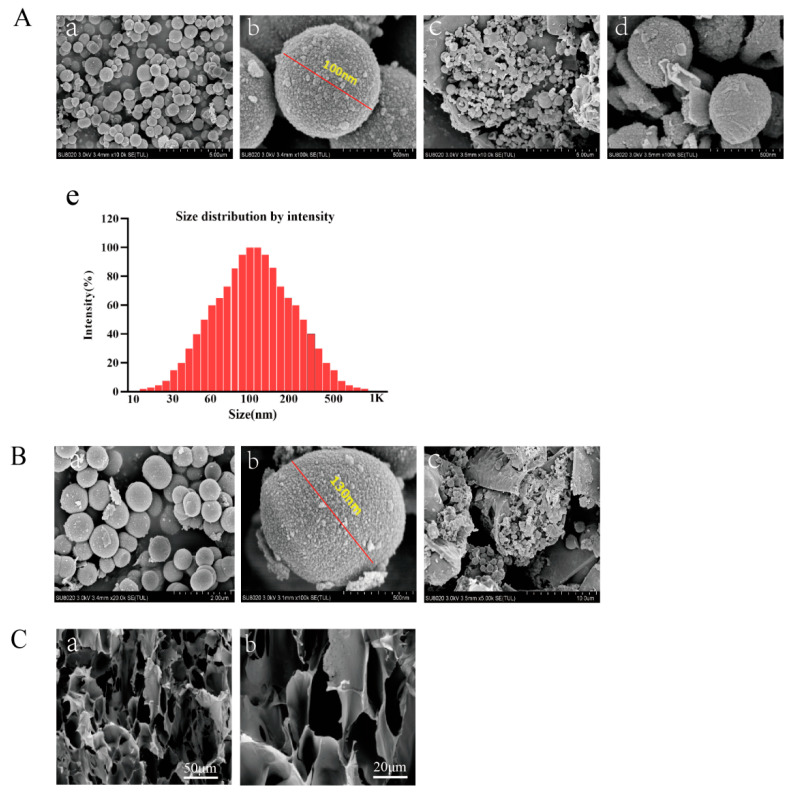
Morphological and structural characterization of the fabricated biomaterials. (**A**) Characterization of DAPT-HAAM. (**a**,**b**) SEM images of spherical DAPT microspheres at different magnifications (scale bars: 5 μm and 500 nm). (**c**,**d**) SEM images confirming the successful enrichment of DAPT microspheres on the decellularized amniotic membrane matrix (scale bars: 5 μm and 500 nm). (**e**) Size distribution by intensity of the DAPT-loaded microspheres. The analysis was performed using dynamic light scattering (DLS), demonstrating a primary peak centered around 100 nm. (**B**) Characterization of VEGF-GG-HA. (**a**,**b**) SEM images of the prepared spherical VEGF microspheres at different magnifications (scale bars: 2 μm and 500 nm). (**c**) SEM image showing the VEGF microspheres incorporated within the GG-HA hydrogel matrix (scale bar: 10 μm). (**C**) Characterization of the final composite material VEGF-GG-HA&DAPT-HAAM. (**a**,**b**) SEM images of the composite at low and high magnification, respectively, revealing its dense and interconnected porous microstructure (scale bars: 50 μm and 20 μm).

**Figure 2 biomedicines-13-02574-f002:**
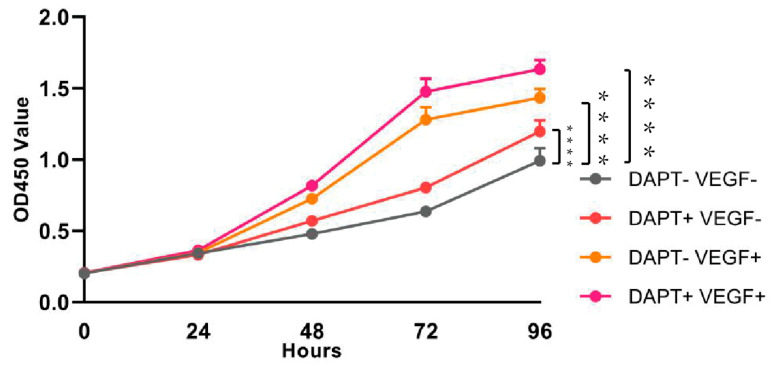
Assessment of cell viability and proliferation by CCK-8 assay. Time-course analysis of THP-1 cell viability from day 1 to day 4 under different treatment conditions: control (no treatment), DAPT-HAAM, VEGF-GG-HA, and the composite VEGF-GG-HA&DAPT-HAAM. Data represent mean ± SD (*n* = 6). **** *p* < 0.0001 vs. control group. DAPT−VEGF− represents the control group (no treatment); DAPT−VEGF+ represents the VEGF-GG-HA group; DAPT+VEGF− represents the DAPT-HAAM group; DAPT+VEGF+ represents the VEGF-GG-HA & DAPT-HAAM group.

**Figure 3 biomedicines-13-02574-f003:**
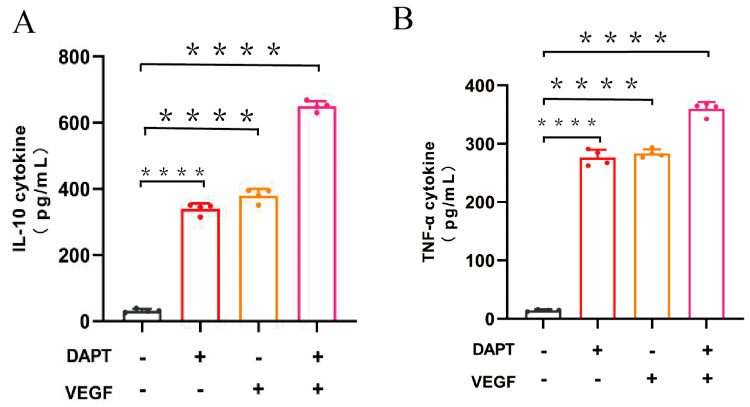
Quantification of cytokine levels by ELISA. (**A**) IL-10 concentration in cell culture supernatant; (**B**) TNF-α concentration in cell culture supernatant. Data represent the mean ± SD (*n* = 4). ****, *p* < 0.0001. DAPT−VEGF− represents the control group (no treatment); DAPT−VEGF+ represents the VEGF-GG-HA group; DAPT+VEGF− represents the DAPT-HAAM group; DAPT+VEGF+ represents the VEGF-GG-HA & DAPT-HAAM group.

**Figure 4 biomedicines-13-02574-f004:**
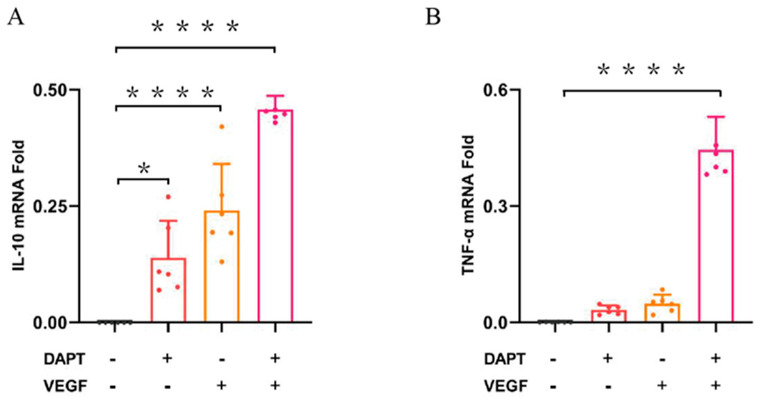
Quantification of cytokine levels by qPCR. (**A**) IL-10 concentration in cell culture supernatant; (**B**) TNF-α concentration in cell culture supernatant. Data represent the mean ± SD (*n* = 6). *, *p* < 0.01, ****, *p* < 0.0001. DAPT−VEGF− represents the control group (no treatment); DAP−VEGF+ represents the VEGF-GG-HA group; DAPT+VEGF−represents the DAPT-HAAM group; DAPT+VEGF+ represents the VEGF-GG-HA & DAPT-HAAM group.

**Figure 5 biomedicines-13-02574-f005:**
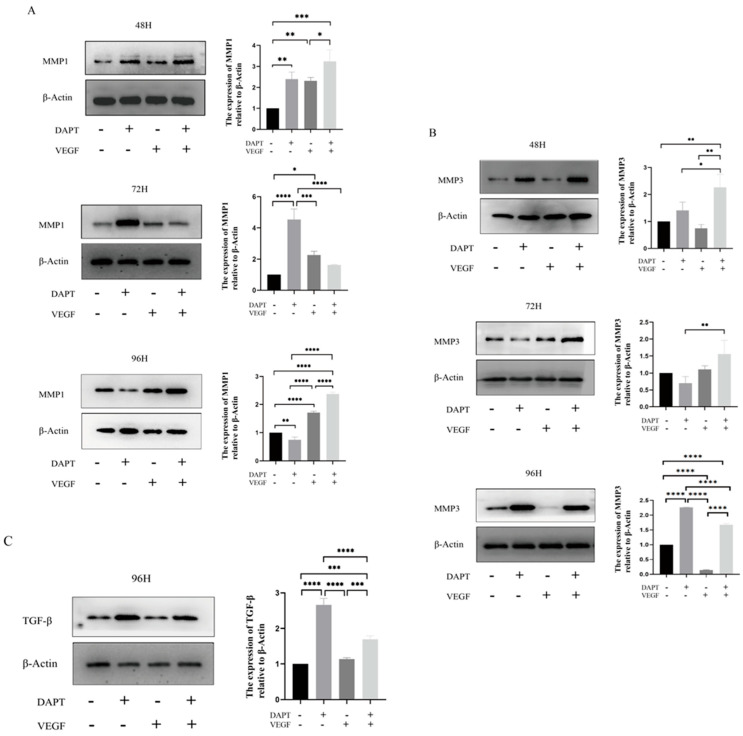
Western blot analysis of protein expression. (**A**) MMP1 protein levels; (**B**) MMP3 protein levels; (**C**) TGF-β protein levels. Data represent the mean ± SD. *, *p* < 0.05, **, *p* < 0.01, ***, *p* < 0.001, ****, *p* < 0.0001. DAPT−VEGF− represents the control group (no treatment); DAPT−VEGF+ represents the VEGF-GG-HA group; DAPT+VEGF− represents the DAPT-HAAM group; DAPT+VEGF+ represents the VEGF-GG-HA & DAPT-HAAM group.

**Figure 6 biomedicines-13-02574-f006:**
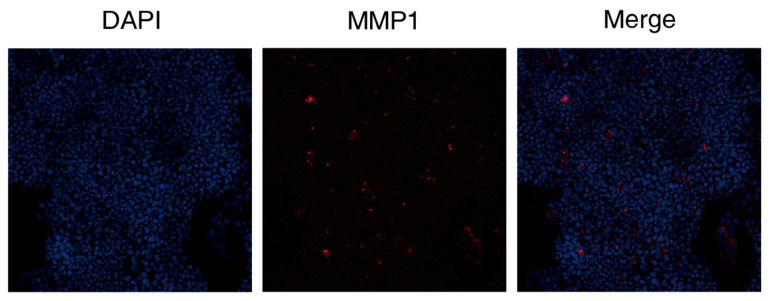
Immunohistochemical analysis of MMP1 expression following DAPT & VEGF treatment. The figure comprises three panels: the DAPI channel (blue, nuclei); the MMP1 immunostaining (red); and the merged image demonstrating the cellular distribution of MMP1.

## Data Availability

All data generated or analyzed during this study are included in this article. Further enquiries can be directed to the corresponding author.

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
