# Peer review of "Immunomodulatory Potential of a Composite Amniotic Membrane Hydrogel for Wound Healing: Effects on Macrophage Cytokine Secretion"

_biomedicines, 2025, doi:10.3390/biomedicines13102574_

Round 1

Reviewer 1 Report

Comments and Suggestions for Authors

The manuscript presents an interesting approach to improving the biological performance of acellular amniotic membrane by combining VEGF-GG-HA hydrogel with DAPT-loaded HAAM. The in vitro findings on macrophage polarization and cytokine modulation are promising for wound healing applications. However, several areas need clarification, deeper analysis, or additional data to strengthen the manuscript.

  1. The manuscript does not provide sufficient data on the hydrogel’s physical/structural characteristics (porosity, swelling capacity, tensile strength, elasticity, degradation rate). Since handling and stability are key limitations of HAAM, please add quantitative measurements or at least discuss these properties in more detail.
  2. The preparation method briefly mentions heating/cooling for crosslinking, but no information is given about the gelation time, thermal stability, or resistance to enzymatic degradation. Please clarify how stable the hydrogel is in simulated physiological conditions.
  3. SEM images confirm spherical VEGF and DAPT microspheres, but it is unclear whether their distribution in the hydrogel and membrane is uniform. Please include data (or representative cross-sectional images) demonstrating microsphere embedding, dispersion, and stability in the hydrogel matrix.
  4. The manuscript states that VEGF and DAPT act synergistically, but the mechanistic explanation remains vague. Please clarify the molecular pathways or present additional evidence supporting their combined effect on macrophage polarization and cytokine release.
  5. Both ELISA and qPCR results show significant increases in TNF-α expression. Since TNF-α is typically pro-inflammatory, explain how this elevation aligns with the claim of promoting M2 polarization and improved healing. Could this reflect a transient stage of inflammation rather than resolution?
  6. Figures require clearer labeling (e.g., axis titles, group names). Some abbreviations such as “DAPT+VEGF+” in Fig. 2 should be explicitly defined in the legend.

Reviewer 2 Report

Comments and Suggestions for Authors

The article "A Modified Acellular Amniotic Membrane Hydrogel Promotes Wound Healing by Regulating Macrophage Polarization" presents research aimed at creating and evaluating a new biomaterial to improve wound healing using a modified acellular amniotic membrane (HAAM), a hyaluronic acid (GG-HA) hydrogel loaded with VEGF and an inhibitor of the Notch signaling pathway (DAPT). Several methods (CCK-8, ELISA, qPCR, Western blot) were used to evaluate the effect of the material on cells. The work combines modern approaches in tissue engineering and immunomodulation to activate macrophages and regulate the inflammatory process in order to accelerate skin regeneration. Below are the comments that will help the authors to finalize the manuscript as follows: 

  1. The text does not contain the abbreviations SDS-PAGE, BSA, PBS. It is recommended to add a full analysis of these terms at their first mention;
  2. Figures 1A and 1B show scanning electron microscopy (SEM) images with the designations a, b, c, however, there are no references or explanations to these letters in the text of the article and under the figure. You should either remove these symbols from the images, or include their description and links in the text and caption to the picture.;
  3. Figure 2 is difficult to understand due to the lack of explanations to the graphs. The figure shows two graphs that visually appear identical. It is necessary to describe the differences between them in more detail in the caption to the figure or in the text, and also add a decoding of the designation of statistical significance (asterisks) so that the reader can correctly interpret the data;
  4. For Figure 3, the caption should be expanded or the drawing itself should be supplemented with elements that will help visually distinguish the two graphs presented under points A and the two graphs under point B. At the moment, the differences between them are not obvious without additional explanations;
  5. The caption to Figure 4 indicates that the analysis was performed using the ELISA method, however, in the relevant part of the article, the data were obtained using quantitative polymerase chain reaction (qPCR). The caption must be adjusted to accurately reflect the analysis method used;
  6. The article does not provide enough detailed statistical data, in particular, there is no clear description of sample sizes and randomization methods. It is recommended to state these aspects more clearly;
  7. The article does not compare the effectiveness of the developed biomaterial with commercial analogues or literary samples. Adding such a comparison would be useful for a more complete understanding of the advantages and practical potential of the new material.

Thus, the article is recommended for publication, but after completion.

Reviewer 3 Report

Comments and Suggestions for Authors

Reviewer Comments

The authors present an interesting study on the development of VEGF-loaded GG-HA hydrogels combined with DAPT-loaded human acellular amniotic membrane (HAAM) to enhance wound healing by modulating macrophage polarization. The concept of combining angiogenic and immunomodulatory agents within bioengineered scaffolds is highly relevant, and the work addresses important limitations of conventional HAAM. The study demonstrates promising results in terms of macrophage modulation, cytokine regulation, and tissue remodeling. However, the manuscript requires major revision as well as clarification from the authors on the following comments.

Comments

  1. In the materials and methods section, DAPT and VEGF microsphere preparation, authors are suggested to provide more detailed information about concentrations, ratios of gelatin to drug, solvent amount used, and mixing conditions in a scientific way.
  2. Describe the Physical characterization (e.g., particle size, encapsulation efficiency, release kinetics) of microspheres with proper references.
  3. Include the Heating, cooling, and sol-gel transition parameters with specific temperatures, durations, and ramping rates.
  4. Authors have used ethylene oxide for sterilization. Please provide the information on how the residues were removed or validated as non-toxic.
  5. For ELISA and qPCR, clarify the number of technical and biological replicates performed. Reporting “n=6” is mentioned for CCK-8 but not consistently across assays.
  6. Details on antibody sources and dilutions should be provided. Currently, only detection kits and membranes are specified.
  7. Chi-square tests are not typically used for quantitative biological assays such as ELISA or qPCR data; one-way ANOVA or t-tests are more appropriate. Please revisit the choice of statistical methods…provide a suitable justification
  8. Clarify abbreviations at first mention (e.g., “RIPA” buffer, “BSA”).
  9. Cytokine Trends (IL-10 vs TNF-α): The results show that both IL-10 (anti-inflammatory, M2-associated) and TNF-α (pro-inflammatory, M1-associated) are elevated by the biomaterials. This dual upregulation is somewhat contradictory when claiming promotion of M2 polarization. The authors should revise the interpretation to explain these findings more cautiously, perhaps by discussing the temporal dynamics of macrophage activation or balancing the effects of DAPT vs VEGF.
  10. In Statistical Presentation, p-values are mentioned (e.g., p < 0.0001), but exact sample sizes (n) for each result are not consistently reported. The authors should revise the Results to include both sample size and effect size (mean ± SD with fold-change values), not just significance markers, to improve transparency.
  11. he authors should also explain why DAPT alone sometimes produced stronger effects than the DAPT+VEGF group, especially for TGF-β.
  12. In Microstructure and Particle Characterization, SEM characterization is limited to particle shape and approximate size. The authors should provide quantitative pore size distribution, porosity percentage, and confirmation of microsphere loading efficiency. Without these quantitative descriptors, claims regarding structural suitability for cell infiltration and bioactivity are incomplete.
  13. The whole results section should be checked for repetitive discussion.

Reviewer 4 Report

Comments and Suggestions for Authors

Dear editor,

I have reviewed the manuscript entitled “A Modified Acellular Amniotic Membrane Hydrogel Promotes 

Wound Healing by Regulating Macrophage Polarization”. My opinion is that the manuscript is well written and therefore is suitable for publication. However, the experimental section needs to be enriched to make it reproducible. Some of my comments and suggestions which could enrich the manuscript are as below:

1- The title is not suitable and should be edited.

2-The abbreviations such as VEGF, GG-HA, DAPT, … should be mentioned in complete form in the first place used.

3- Provide a nomenclature.

4- Provide information on THP-1 cells and the reason of using it.

5- Briefly introduce the HAAM production process.

6- The experimental section needs to be rewritten in a way that researchers can be able to repeat the work. For example, the amounts of material used, the heating temperature and time of stirring needs to be provided.

7- Provide information on the DAPT powder. The amount used should also be mentioned in the text.

8- Provide information on the limonene receiving solution.

9- What do you mean by the lyophilization agent? “The mixture was freeze-dried using a lyophilization agent” freeze drying is a lyophilization process.

10- Provide information on: “sterilized with ethylene oxide”.

11- It has been mentioned that “The assembly was heated to the sol-gel transition temperature”. How has this temperature been obtained and mention the temperature in the text.

12- CO2 should be CO2

13- All of the protocols used should be mentioned and referenced in the text.

14- All of the used material such as HA, Gelzan gel powder, … should be introduced. Provide the manufacture, code, purity and etc.

15- All the prepared material such as VEGF-GG-HA needs to be proved that they have been successfully obtained. It just has been supposed that the reactions are successful. Provide analysis such as FTIR and … to prove to existence of all the material produced.

16- Provide images of the hydrogels and films which has been obtained?

17- The reason of using the analysis and staining’s should be mentioned in the appropriate sections to be more informative for the readers.

18- For the DAPT microspheres provide the particle size distribution and mean particle size with error bars.

19- Please explain for the readers that what happens during the conversion of macrophage recruitment from M1-type to M2-type.

20- What about Phalloidin staining can it be useful?

21- Provide some detailed information in the text on the reactions occurring during the processes and molecular structures if possible.

Round 2

Reviewer 2 Report

Comments and Suggestions for Authors

The authors have corrected all the comments, the article can be published

Reviewer 3 Report

Comments and Suggestions for Authors

The manuscript has been revised to the satisfactory level by the authors.. All the comments are addressed well.  So it can be considered for publication.

Reviewer 4 Report

Comments and Suggestions for Authors

The mauscript is acceptable in present form